# Reference-based MRI Reconstruction Using Texture Transformer

**Pengfei Guo**                                                                        PGUO4@JHU.EDU
**Vishal M. Patel**                                                                  VPATEL36@JHU.EDU
*Johns Hopkins University, Baltimore, MD 21218, USA*

**Editors:** Accepted for publication at MIDL 2023

## Abstract

Deep Learning (DL) based methods for magnetic resonance (MR) image reconstruction have been shown to produce superior performance. However, previous methods either only leverage under-sampled data or require a paired fully-sampled auxiliary MR sequence to perform the guidance-based reconstruction. Consequently, existing approaches neglect to explore attention mechanisms that can transfer texture from reference data to under-sampled data within a single MR sequence, which either limits the performance of these approaches or increases the difficulty of data acquisition. In this paper, we propose a novel **T**exture **T**ransformer **M**odule (**TTM**) for the reference-based MR image reconstruction. The TTM facilitates joint feature learning across under-sampled and reference data, so feature correspondences can be discovered by attention and accurate texture features can be leveraged during reconstruction. Notably, TTM can be stacked on prior MRI reconstruction methods to improve their performance. In addition, a **R**ecurrent **T**ransformer **R**econstruction backbone (**RTR**) is proposed to further improve the performance in a unified framework. Extensive experiments demonstrate the effectiveness of TTM and show that RTR can achieve prominent results on multiple datasets.

**Keywords:** MRI Reconstruction, Deep Learning, Transformer.

## 1. Introduction

Magnetic resonance imaging (MRI) is one of the most widely used noninvasive medical imaging techniques. However, due to the acquisition hardware limitation, the relatively slow data acquisition process of MRI impedes its development in many clinical applications. In addition to the exploitation of advanced hardware and parallel imaging (Pruessmann et al., 1999; Taouli et al., 2004; Niendorf and Sodickson, 2006), a common approach that can shorten the image acquisition time is Compressed Sensing (CS) (Lustig et al., 2008) where MR images are reconstructed from under-sampled $k$-space data. Though CS reconstruction methods are able to recover images, they lack the ability to recover noise-like textures and introduce high-frequency oscillatory artifacts when large errors are not properly reduced during optimization (Ravishankar and Bresler, 2010).

Recently, DL-based methods have significantly improved the quality of recovered images in MRI reconstruction. Previous approaches are essentially based on two paradigms – single image reconstruction (SIR), and guidance-based image reconstruction (GIR). The SIR problem aims to recover a fully-sampled image from a single under-sampled $k$-space data, as shown in Fig. 1(a). Several SIR methods (Ronneberger et al., 2015; Eo et al., 2018; Schlemper et al., 2017; Wang et al., 2019; Guo et al., 2021a; Yiasemis et al., 2022) have

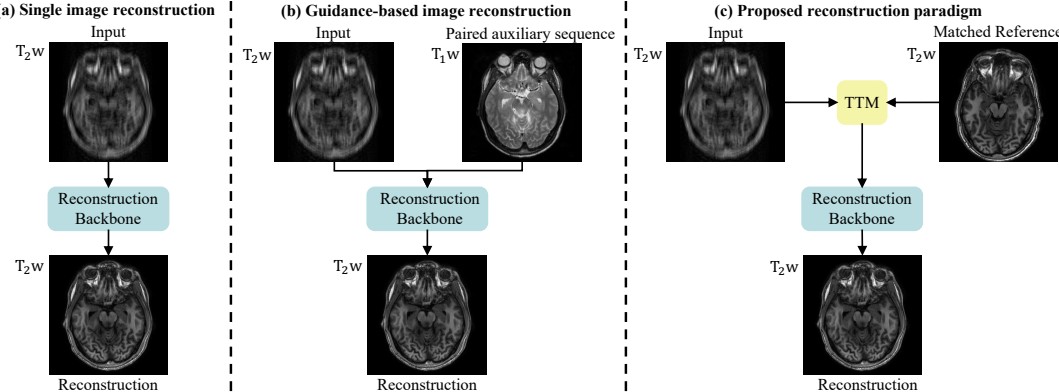

Figure 1: The schematics of different MRI reconstruction paradigms. It is worth noting that the matched reference in the proposed paradigm can be from another subject.

demonstrated superior performance over the CS-based methods. However, in challenging cases, SIR methods often result in blurry artifacts, since the anatomic textures have been excessively degraded by the $k$-space under-sampling process. Several methods (Dar et al., 2020; Guo et al., 2020; Wang et al., 2020; Guo et al., 2021b; Zhou and Zhou, 2020) have focused on GIR, which explores feature correspondences from a co-registered (paired) auxiliary MR sequence to produce visually pleasing results. However, those approaches require a multi-modal dataset for training. For example, as shown in Fig. 1(b), reconstructing a $T_2$-weighted image requires a co-registered fully-sampled $T_1$-weighted image (Xiang et al., 2018; Souza et al., 2020; Beauferris et al., 2022) or co-registered under-sampled auxiliary sequences (Peng et al., 2020) from the same subject as the reference, which significantly increases the difficulty of data acquisition and impedes the practicality of GIR methods.

To address these issues and inspired by the success of reference-based image restoration tasks (Yang et al., 2020; Vaswani et al., 2017), we propose a novel Texture Transformer Module (TTM) for MR image reconstruction. Specifically, we formulate the extracted features from the under-sampled image and the matched reference image as the query and key in a transformer. The produced attention maps are used to transfer the features from the reference image into the features extracted from the under-sampled data. As shown in Fig. 1(c), instead of relying on a co-registered (paired) auxiliary sequence as required by previous GIR methods, reconstructing a $T_2$-weighted image in the proposed paradigm only demands a matched fully-sampled $T_2$-weighted image from the reference data split, which can considerably alleviate the difficulty of data acquisition in the case where subjects do not have paired auxiliary MR sequences. It is worth noting that exploring feature correspondences between different subjects is non-trivial, since human anatomy shares common radiographic patterns (Giedd et al., 2009). To the best of our knowledge, the proposed TTM is the first attempt to explore the relevance and transfer of texture features within a single MR sequence using a transformer.

In summary, the following are our key contributions: **1.** A new reference-based MRI reconstruction paradigm is formulated and the proposed TTM introduces a powerful way

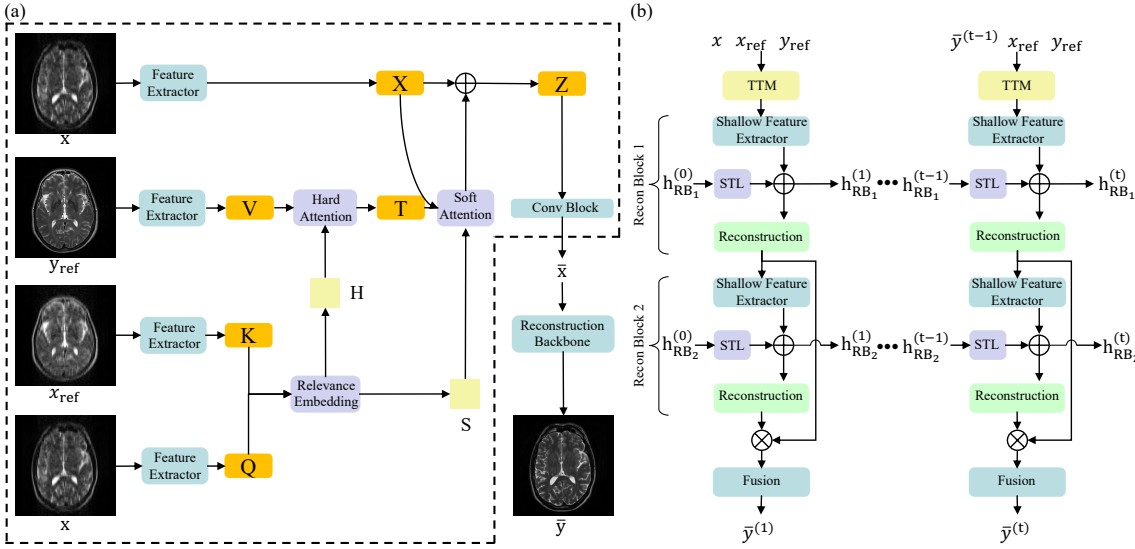

Figure 2: (a) An overview of the proposed texture transformer module. (b) A schematic of unrolled iterations of the proposed RTR. STL consists of the Swin Transformer Layers.

to promote joint feature learning across under-sampled and reference data within a single MR sequence. **2.** A recurrent transformer model, namely RTR, is further proposed for improving the reconstruction fidelity. **3.** Experiments on multiple datasets with different acceleration factors show that TTM can boost the performance of existing backbones in the proposed paradigm and the superiority of RTR in MRI reconstruction.

## 2. Methodology

Accelerated MRI reconstruction is an inverse problem in which the objective is to reconstruct a fully-sampled image from under-sampled $k$-space data. The data acquisition process (Schlemper et al., 2017) can be formulated as follows:

$$x' = F_D y + \epsilon, \tag{1}$$

where $x'$ denotes the observed under-sampled k-space, $y$ is the fully-sampled image, and $\epsilon$ denotes the noise. $F$ and $F^{-1}$ are the Fourier transform matrix and its inverse, respectively. We denote $x = F^{-1}(x')$ as the observed under-sampled image. $F_D$ represents the under-sampling Fourier encoding matrix that is defined as the multiplication of the Fourier transform matrix $F$ with a binary undersampling mask matrix $D$. The goal is to estimate $y$ from the observed under-sampled k-space data $x'$.

### 2.1. Texture Transformer Module

Fig. 2(a) shows an overview of the proposed texture transformer module. $x$, $x_{\text{ref}}$, and $y_{\text{ref}}$ represent the input under-sampled image, the under-sampled reference image, and the

fully-sampled reference image, respectively. Since we apply the same under-sampling operations, $x_{\text{ref}}$ is domain-consistent with $x$. The proposed TTM consists of five components as follows: a feature extractor, a relevance embedding module, a hard attention module for feature transfer, a soft attention module for feature synthesis, and a convolutional block for generating output. TTM takes $x$, $x_{\text{ref}}$, and $y_{\text{ref}}$ as input, and outputs a synthesized under-sampled image $\bar{x}$, which contains real and imaginary channels and can be further used to generate the fully-sampled prediction $\bar{y}$ by a reconstruction backbone. In what follows, we describe different parts of TTM in detail.

**Feature Extractor.** Extracting accurate and proper texture information plays an essential role to facilitate the downstream fully-sampled MR image reconstruction. To encourage joint feature learning, we introduce a learnable feature extractor whose parameters are updated during training. Query ($Q$), Key ($K$), and Value ($V$) are three basic elements of the attention mechanism in a transformer (Vaswani et al., 2017), which are formulated as follows:

$$Q = FE(x), \quad K = FE(x_{\text{ref}}), \quad V = FE(y_{\text{ref}}), \tag{2}$$

where $FE(\cdot)$ denotes the output of feature extractor. The proposed feature extractor consists of 4 convolutional blocks. To enable high flexibility in expressing the structure of input data, the channel dimension of the output of the feature extractor is set to 64.

**Relevance Embedding Module.** To embed the relevance between $x$ and $x_{\text{ref}}$, the similarity between $Q$ and $K$ is estimated by the relevance embedding module. $Q$ and $K$ are unfolded into patches with size of $16 \times 16$ and we denote them as $q_i$ and $k_j$, where $i$ and $j$ indicate the spatial locations of patches. For each patch pair between $Q$ and $K$, we can calculate the relevance score $r_{i,j}$ by the normalized inner product as follows:

$$r_{i,j} = \left\langle \frac{q_i}{\|q_i\|}, \frac{k_j}{\|k_j\|} \right\rangle. \tag{3}$$

The obtained relevance scores are used to generate hard and soft attention maps.

**Hard Attention Module.** Hard attention module is designed to search the most relevant fully-sampled texture features from $V$. Rather than taking a weighted sum of $V$ for each query $q_i$, only the most relevant feature in $V$ is transferred for the corresponding query $q_i$. Such design can prevent blurry outputs and promote the accuracy of transferring full-sampled texture features (Yang et al., 2020). Specifically, the $i$-th element $h_i$ of the hard-attention map $H$ can be calculated as follows:

$$h_i = \operatorname*{argmax}_j r_{i,j}. \tag{4}$$

Each $h_i$ can be treated as a hard index which indicates the most relevant position in $y_{\text{ref}}$ to the $i$-th position in $x$. Also, $V = FE(y_{\text{ref}})$ is unfolded into patches with size of $16 \times 16$ and the transferred texture features $T$ from $y_{\text{ref}}$ are selected by using $H$ as the indices in Eq. 4. Let $t_i$ denote the patch of $T$ in the $i$-th position. Then, $t_i$ is selected from the $h_i$-th position of $V$ and can be expressed as $t_i = V_{h_i}$.

**Soft Attention Module.** Soft attention module aims to generate the synthesized features $Z$ using the transferred texture features $T$ and extracted features $X$ from the input under-sampled image $x$. To take the confidence of the transferred texture features into account

during the synthesis process, a soft attention map $S$ is calculated as follows:

$$s_i = \max_j r_{i,j}, \tag{5}$$

where $s_i$ is the $i$-th position of the soft-attention map $S$ which represents the confidence of the transferred texture features in this position. To avoid losing information from the input image, $T$, and $X$ are first fused, then the fused features are element-wise multiplied by the soft attention map $S$ and added back to $X$ to produce the synthesized features $Z$. This synthesis process can be expressed as follows:

$$Z = X + \text{Conv}(X \otimes T) \odot S, \tag{6}$$

where Conv represents a convolutional layer. $\otimes$ and $\odot$ denote the channel-wise concatenation and the element-wise multiplication, respectively. Finally, the channel dimension of the synthesized features $Z$ is reduced to 2 (*i.e.*, real and imaginary channels) by a convolutional block. The synthesized under-sampled image $\bar{x}$ containing the transferred texture features can be further used as the input of image reconstruction backbones. We use *mutual information* as the metric to create input-reference matches. More details of this matching process can be found in Appendix A.

## 2.2. Recurrent Transformer for MRI Reconstruction

Inspired by previous iterative MRI reconstruction approaches (Guo et al., 2021a; Qin et al., 2018; Wang et al., 2019) and recent advancements in transformer models (Liang et al., 2021; Liu et al., 2021), we propose a novel recurrent transformer model RTR for reference-based MRI reconstruction. Figure 2(b) shows the unrolled RTR iterations. RTR consists of a texture transformer module (TTM), two reconstruction blocks (RBs), and a fusion module (FM). Each RB contains a shallow feature extractor (SFE), Swin transformer layers (STL) (Liu et al., 2021) for deep feature extraction, and a reconstruction module (RM). SFE, RM, and FM are built based on convolution layers for efficient learning and stable optimization (Xiao et al., 2021). The last layer of RM is a data consistency layer that produces a linear combination between the network prediction and the original measurement for reinforcing the data consistency in the $k$-space (Schlemper et al., 2017). To avoid losing high-resolution information during reconstitution, two RBs map input to the original dimension but perform different down-sampling operations (*i.e.*, $\times 2$ and $\times 1$) controlled by SFE for multi-scale perception. Let $\text{RB}_i$ denote the $i$-th reconstruction block. At each time step $t$, the hidden state $h_{\text{RB}_i}^{(t)}$ in $\text{RB}_i$ is updated recursively as follows:

$$h_{\text{RB}_i}^{(t)} = SFE(\bar{x}^{(t)}) + STL(h_{\text{RB}_i}^{(t-1)}). \tag{7}$$

Then reconstruction module takes the updated hidden state to produce the output of $\text{RB}_i$. The workflow of RTR in the $t$-th iteration can be described as follows:

$$
\begin{aligned}
\bar{x}^{(t)} &= TTM(\bar{y}^{(t-1)}, x_{\text{ref}}, y_{\text{ref}}), \\
\bar{y}_{\text{RB}_1}^{(t)} &= RB_1(\bar{x}^{(t)}, h_{\text{RB}_1}^{(t-1)}), \\
\bar{y}_{\text{RB}_2}^{(t)} &= RB_2(\bar{y}_{\text{RB}_1}^{(t)}, h_{\text{RB}_2}^{(t-1)}), \\
\bar{y}^{(t)} &= FM(\bar{y}_{\text{RB}_1}^{(t)} \otimes \bar{y}_{\text{RB}_2}^{(t)}),
\end{aligned}
\tag{8}
$$

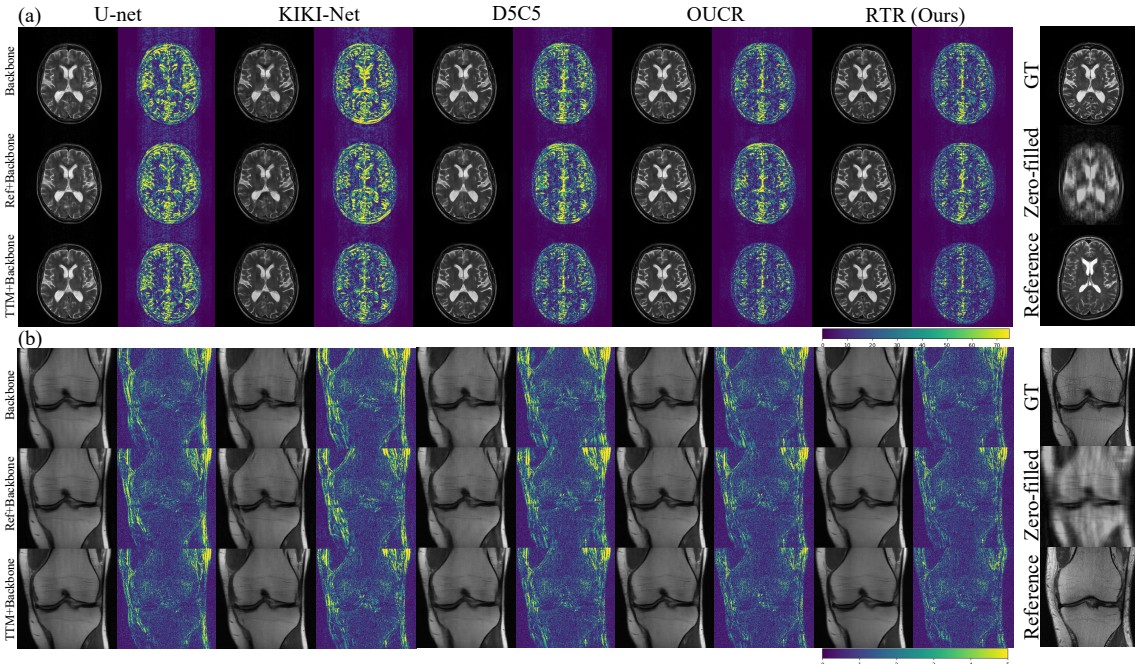

Figure 3: Visual comparison of different MRI reconstruction methods (AF=8) on (a) IXI and (b) fastMRI among 3 strategies. The second column of each sub-figure shows the absolute difference between the reconstructed image and the ground truth.

where $\otimes$ denotes the channel-wise concatenation. FM fuses outputs of two reconstruction blocks to produce the intermediate reconstruction result $\bar{y}^{(t)}$, which also serves as the input for the next recurrent iteration. As shown in Eq. 8, the proposed RTR is also able to perform conventional single-image reconstruction by simply removing the stacked TTM. The detailed network configuration can be found in Appendix B.

## 3. Experiments and Results

**Dataset.** Experiments are conducted on the IXI (brain development.org, 2015) and fastMRI (Knoll et al., 2020) datasets. 135 $T_2$-weighted axial brain MRI scans from IXI are analyzed. The training, validation, reference, and testing data splits are as 100/10/10/15. Each scan approximately provides 130 images. Since IXI (brain development.org, 2015) does not provide raw k-space data, we follow common practices (Peng et al., 2020) to generate simulated k-space data for a proof-of-concept study. The fastMRI dataset contains 1,172 complex-valued single-coil coronal proton density (PD)-weighted knee MRI scans. The training, reference, and testing data splits are as 923/50/199. Each scan approximately provides 35 knee images. Since these two datasets originally lack the reference data split, we randomly sample scans as the reference data.

**Implementation Details.** All models were trained using the $\ell_1$ loss that is enforced at each with Adam optimizer based on the following hyperparameters: initial learning rate

Table 1: PSNR/SSIM comparisons among different strategies. AVG. Gain represents the performance changes compared to the Backbone. We report the mean and standard deviation across test subjects.

| Strategy | Reconstruction Backbone | | | | | AVG. Gain |
|---|---|---|---|---|---|---|
| **IXI (AF=4)** | | | | | | |
| | U-net | KIKI-Net | D5C5 | OUCR | RTR (Ours) | |
| Backbone | 33.49/91.31 | 34.73/94.56 | 36.61/96.56 | 37.41/97.36 | 38.16/97.67 | - |
| Ref+Backbone | 33.52/91.34 | 34.70/94.38 | 36.38/96.24 | 37.40/97.35 | 37.89/97.56 | − 0.10/− 0.12 |
| TTM+Backbone | 34.25/92.38 | 36.91/96.82 | 38.23/97.44 | 38.22/97.72 | **38.90/97.99** | + 1.22/+ 0.98 |
| **IXI (AF=8)** | | | | | | |
| | U-net | KIKI-Net | D5C5 | OUCR | RTR (Ours) | |
| Backbone | 28.65/84.61 | 28.75/86.28 | 29.35/88.38 | 30.40/90.89 | 30.97/91.87 | - |
| Ref+Backbone | 28.61/84.47 | 28.77/86.52 | 29.29/88.99 | 30.22/90.62 | 30.69/91.53 | − 0.11/+ 0.03 |
| TTM+Backbone | 29.51/86.36 | 30.21/90.10 | 30.70/91.60 | 31.38/92.42 | **32.39/93.63** | + 1.21/+ 2.42 |
| **fastMRI (AF=4)** | | | | | | |
| | U-net | KIKI-Net | D5C5 | OUCR | RTR (Ours) | |
| Backbone | 31.78/71.34 | 31.92/71.80 | 32.18/72.34 | 32.37/72.82 | 32.48/73.31 | - |
| Ref+Backbone | 31.81/71.39 | 31.94/71.80 | 32.14/72.38 | 32.34/72.78 | 32.47/73.23 | − 0.01/− 0.01 |
| TTM+Backbone | 32.02/71.88 | 32.16/72.38 | 32.35/72.71 | 32.52/73.36 | **32.61/73.46** | + 0.19/+ 0.44 |
| **fastMRI (AF=8)** | | | | | | |
| | U-net | KIKI-Net | D5C5 | OUCR | RTR (Ours) | |
| Backbone | 29.65/63.93 | 29.26/63.87 | 29.57/64.32 | 30.07/65.35 | 30.40/66.15 | - |
| Ref+Backbone | 29.66/63.51 | 29.27/63.77 | 29.58/64.35 | 30.10/65.39 | 30.33/65.85 | − 0.01/− 0.15 |
| TTM+Backbone | 29.98/64.81 | 29.70/64.71 | 30.05/65.31 | 30.39/66.05 | **30.72/0.6673** | + 0.38/+ 0.80 |

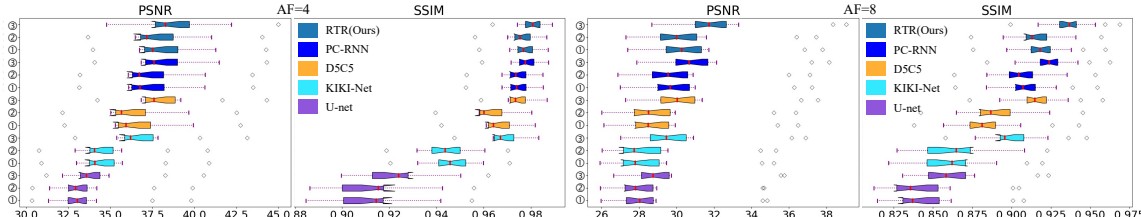

Figure 4: Boxplots of reconstruction performance on IXI among five methods. ①, ②, and ③ represent Backbone, Ref+Backbone, and TTM+Backbone, respectively.

of $1.5 \times 10^{-4}$ then reduced by a factor of 0.9 every 5 epochs; 50 maximum epochs; batch size of 8; the number of recurrent iterations $T$ of 3 (if applicable). The sampling mask function with $4\times$ and $8\times$ accelerations is used for simulating the complex-valued $k$-space measurements. Structural Similarity Index Measure (SSIM) and Peak Signal-to-Noise Ratio (PSNR) are used as evaluation metrics for comparisons. We implement the proposed model using PyTorch on Nvidia RTX8000 GPUs.

**Reconstruction Results.** To evaluate the effectiveness of the proposed TTM, we observe the performance change of adding TTM to five MRI reconstruction methods, including U-net (Ronneberger et al., 2015), KIKI-Net (Eo et al., 2018), D5C5 (Schlemper et al., 2017), OUCR (Guo et al., 2021a), and the proposed RTR. For a fair comparison, U-net (Ronneberger et al., 2015) is modified for data with real and imaginary channels and a data

consistency (DC) layer is added at the end of the network. Table 1 shows the results corresponding to five reconstruction methods evaluated on different acceleration factors (AF). We first compare the performance of different methods without any modifications and denote them as **Backbone**. It is also possible to naively leverage information provided by the reference images. In this case, we just concatenate the reference image with the input as additional channels. It is worth noting that there are no registration steps involved in this setting. The results corresponding to this strategy are shown in Table 1 under the label **Ref+Backbone**. It is worth noting that TTM is removed from our proposed RTR for fair comparisons in the above two scenarios. Finally, we can obtain models that use the proposed TTM to transfer texture information to facilitate reconstruction backbones. These experiments are denoted as **TTM+Backbone**.

From Table 1, we can make the following observations: (i) The proposed RTR achieves the best performance across three training strategies for all acceleration factors. The superior performance of RTR demonstrates the merit of the proposed transformer architecture. (ii) Naively treating the reference image as an additional input cannot improve the reconstruction performance. Due to the lack of an efficient mechanism for searching and transferring the texture information in the reference images, it results in slightly lower performance across four different methods. For example, the AVG. Gain of the Ref+Backbone strategy is $-0.10$ dB and $-0.11$ dB on IXI for AF=4 and AF=8, respectively. (iii) By using the proposed TTM, we are able to achieve obvious improvements as compared to the models that naively use the reference images and the original backbones. As shown in Table 1, the AVG. Gain is about 1.2 dB in experiments of two accelerations on IXI. (iv) The effectiveness of TTM is consistent across different reconstruction methods, acceleration factors, and datasets. Even for the powerful iterative methods (*i.e.*, OUCR (Guo et al., 2021a) and our proposed RTR), the proposed TTM still can improve the reconstruction quality by 0.32 dB on fastMRI for AF=8 (the most challenging scenario). All reported improvements achieved by RTR are statistically significant ($p < 0.05$) as shown in Appendix C.

Fig. 3 shows the qualitative performance of different methods. As can be seen from the error maps, the proposed RTR yields the reconstructed images with remarkable visual similarity to the GT images compared to the other methods across the two datasets. Meanwhile, all models that make use of TTM exhibit a better ability to suppress the overall errors (see error maps in Figure 3 and zoomed-in images in Appendix E), which is consistent with our quantitative results. Fig. 4 shows the boxplots of reconstruction quality in different strategies. We can observe that the overall spread of metrics is shifted to the right in TTM+Backbone, which implies the effectiveness of our proposed TTM for most test cases.

Table 2: Ablation study of proposed modules in TTM.

| Method | HA | SA | PSNR/SSIM |
|---|---|---|---|
| Base | | | 33.49/0.9131 |
| SA+Base | | ✓ | 33.86/0.9156 |
| HA+Base | ✓ | | 34.07/0.9172 |
| TTM+Base | ✓ | ✓ | 34.25/0.9238 |

Table 3: Ablation study of proposed modules in RTR.

| Method | RB$_1$ | RB$_2$ | FM | TTM | PSNR/SSIM |
|---|---|---|---|---|---|
| RB$_1$ | ✓ | | | | 36.02/0.9589 |
| RB$_1$ +RB$_2$ | ✓ | ✓ | | | 37.38/0.9707 |
| RTR | ✓ | ✓ | ✓ | | 38.16/0.9764 |
| TTM+RTR | ✓ | ✓ | ✓ | ✓ | 38.90/0.9799 |

**Ablation Study.** The contribution of the proposed TTM is demonstrated by comparisons between three training strategies in Table 1. Moreover, we conduct a detailed ablation study

on IXI (AF=4) to analyze the effectiveness of different components in the proposed TTM. In this case, we use U-net (Ronneberger et al., 2015) as our base model. On top of the base model, we progressively add a hard attention module (HA) and a soft attention module (SA). Ablation results are shown in Table 2. As one can see, both HA and SA show positive contributions in terms of the reconstruction quality. After adding SA, relevant texture features are enhanced and the less relevant ones are suppressed during feature synthesis. When HA is added, PSNR is improved from 33.49dB to 34.07dB, which demonstrates the effectiveness of HA in feature transformation. It is worth noting that the structure of TTM is compact, so it only introduces a few extra trainable parameters. In addition, Table 3 shows the ablation study of the proposed modules in RTR. The ablation study regarding the robustness to reference images of different similarity and additional visualizations can be found in Appendix D.

## 4. Conclusion

In this paper, we proposed a novel TTM for the MR image reconstruction task which is able to transfer texture features from the reference image to the under-sampled image. Unlike previous guidance-based MRI reconstruction methods, TTM is able to explore the reference images within a single MR sequence and does not require paired auxiliary MR sequences. Moreover, the proposed TTM can be directly stacked on the existing MRI reconstruction methods to further improve their performance. In addition, RTR is further proposed to reconstruct high-quality MR images in a unified framework. Recurrent transformer-based design in RTR leads to outstanding performance in both single-image reconstruction and the proposed reference-based reconstruction paradigms. Experimental results demonstrate the effectiveness of the proposed TTM in multiple MRI reconstruction approaches as well as the promising potential of applying transformer-based models in the MRI reconstruction task. While our proposed method yielded a competitive performance, we include the discussion about limitations in Appendix F.

## Acknowledgments

This work was supported by an ARO grant W911NF-21-1-0135.

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

## Appendix A. Mutual Information-Based Matching.

We match input under-sampled image $x$ with a corresponding under-sampled reference image $x_{\text{ref}}$ and consequently find the fully-sampled reference image $y_{\text{ref}}$ from the reference data split during data preprocessing. It is important to accurately identify the matched slices between under-sampled images and the reference data in the proposed paradigm. We exploit a simple yet effective mutual information(MI)-based slice matching algorithm that can find the closest corresponding slice in reference data.

Let $\mathcal{X}_{\text{ref}} = \{x_{\text{ref}}^k | i = 1, 2, \ldots, n\}$ be a set that contains n under-sampled slices of reference data, where $x_{\text{ref}}^k = F^{-1}(F_D y_{\text{ref}}^k + \epsilon)$ and $x = F^{-1}(F_D y + \epsilon)$ have the same under-sampling operation $F_D$. The MI between $x$ and $x_{\text{ref}}^k$ can be calculated as follows:

$$\text{MI}_k = H(x) + H(x_{\text{ref}}^k) - H(x, x_{\text{ref}}^k), \tag{9}$$

where $H(x)$ is the individual entropy of $x$ and $H(x, x_{\text{ref}}^k)$ is the joint entropy between $x$ and $x_{\text{ref}}^k$. The matched index indicates the most relevant slice in reference data can be found by solving:

$$\underset{k}{\text{argmax}}\ \text{MI}_k. \tag{10}$$

# Appendix B.  Additional Implementation Details.

Table 4: Configuration of TTM. TTM only consists of 0.7M trainable parameters, which are negligible compared to reconstruction backbones.

| Block | Layer | In Chanel | Out Chanel | Kernel size | Stride | Padding |
|-------|-------|-----------|------------|-------------|--------|---------|
| Feature Extractor | Conv | 2 | 64 | 3 | 1 | 1 |
| | ReLu | 64 | 64 | - | - | - |
| | Conv | 64 | 64 | 3 | 1 | 1 |
| | ReLu | 64 | 64 | - | - | - |
| | Conv | 64 | 64 | 3 | 1 | 1 |
| | ReLu | 64 | 64 | - | - | - |
| Conv Block | ResBlock | 64 | 64 | 3 | 1 | 1 |
| | ResBlock | 64 | 64 | 3 | 1 | 1 |
| | ResBlock | 64 | 64 | 3 | 1 | 1 |
| | ResBlock | 64 | 2 | 3 | 1 | 1 |

Table 5: Configuration of RTR. Swin is a Swin Transformer Layer.

| Module | Block | Layer | kernel size | Stride | Padding | In chanels | Out chanels |
|--------|-------|-------|-------------|--------|---------|------------|-------------|
| $RB_1$ | SFE | Conv | 3 | 1 | 1 | 2 | 96 |
| | | ReLu | - | - | - | 96 | 96 |
| | | Conv | 4 | 2 | 1 | 96 | 96 |
| | STL | Swin | - | - | - | 96 | 96 |
| | | Swin | - | - | - | 96 | 96 |
| | | Swin | - | - | - | 96 | 96 |
| | | Swin | - | - | - | 96 | 96 |
| | RM | TransposeConv | 4 | 2 | 1 | 96 | 96 |
| | | ReLu | - | - | - | 96 | 96 |
| | | TransposeConv | 3 | 1 | 1 | 96 | 2 |
| $RB_2$ | SFE | Conv | 3 | 1 | 1 | 2 | 48 |
| | | ReLu | - | - | - | 48 | 48 |
| | | Conv | 3 | 1 | 1 | 48 | 48 |
| | STL | Swin | - | - | - | 48 | 48 |
| | | Swin | - | - | - | 48 | 48 |
| | RM | TransposeConv | 3 | 1 | 1 | 48 | 48 |
| | | ReLu | - | - | - | 48 | 48 |
| | | TransposeConv | 3 | 1 | 1 | 48 | 2 |

## Appendix C. Statistical Significance Tests.

Table 6: The $p$ values of statistical significance among different methods. To investigate the performance improvement of the proposed RTR, we conduct paired t-test based on the PSNR of the reconstructed images between RTR and other methods.

| Strategy | IXI (AF=4) | | | | | IXI (AF=8) | | | | |
|---|---|---|---|---|---|---|---|---|---|---|
| | U-net | KIKI-Net | D5C5 | OUCR | RTR | U-net | KIKI-Net | D5C5 | OUCR | RTR |
| Backbone | 5.73e-14 | 2.72e-14 | 4.01e-15 | 3.19e-15 | - | 7.56e-11 | 2.11e-11 | 2.02e-14 | 3.77e-12 | - |
| Ref+Backbone | 9.73e-14 | 2.48e-14 | 2.53e-16 | 1.88e-13 | - | 1.83e-10 | 7.52e-12 | 2.03e-14 | 1.85e-11 | - |
| TTM+Backbone | 2.68e-14 | 2.09e-14 | 7.28e-13 | 1.38e-15 | - | 1.87e-12 | 1.41e-14 | 2.99e-16 | 4.62e-16 | - |
| Strategy | fastMRI (AF=4) | | | | | fastMRI (AF=8) | | | | |
| | U-net | KIKI-Net | D5C5 | OUCR | RTR | U-net | KIKI-Net | D5C5 | OUCR | RTR |
| Backbone | 1.81e-68 | 8.42e-59 | 2.21e-55 | 6.63e-83 | - | 8.65e-62 | 2.38e-50 | 9.55e-50 | 8.91e-52 | - |
| Ref+Backbone | 1.28e-69 | 3.01e-56 | 6.88e-61 | 1.05e-77 | - | 1.16e-67 | 1.74e-51 | 1.73e-52 | 1.60e-46 | - |
| TTM+Backbone | 3.19e-62 | 2.05e-50 | 2.20e-48 | 1.35e-39 | - | 1.99-57 | 7.25e-47 | 1.45e-50 | 2.69e-51 | - |

## Appendix D. Additional Results.

Table 7: Ablation study about reference images of different similarity on IXI (AF=4). "random" means that we randomly pick an image from the reference data split. "noise" means that we generate a random noise image as the reference. Oracle indicates the models without TTM. "Abnormal" means that we replace the original reference data split with the data from in-house post-surgery patients with malignant glioma. Since human anatomy shares a lot of common patterns, even with randomly picked full-sampled reference data, TTM can still utilize these common patterns and improve the reconstruction performance.

| Level | Similarity | Methods | |
|---|---|---|---|
| | | TTM+OUCR | TTM+RTR |
| L1 | $\underset{k}{\operatorname{argmax}} \operatorname{MI}_k$ | 38.22 | 38.90 |
| L2 | random | 38.14 | 38.88 |
| L3 | $\underset{k}{\operatorname{argmin}} \operatorname{MI}_k$ | 38.01 | 38.79 |
| L4 | noise | 37.39 | 38.01 |
| Abnormal | $\underset{k}{\operatorname{argmax}} \operatorname{MI}_k$ | 38.11 | 38.85 |
| Oracle | - | 37.41 | 38.16 |

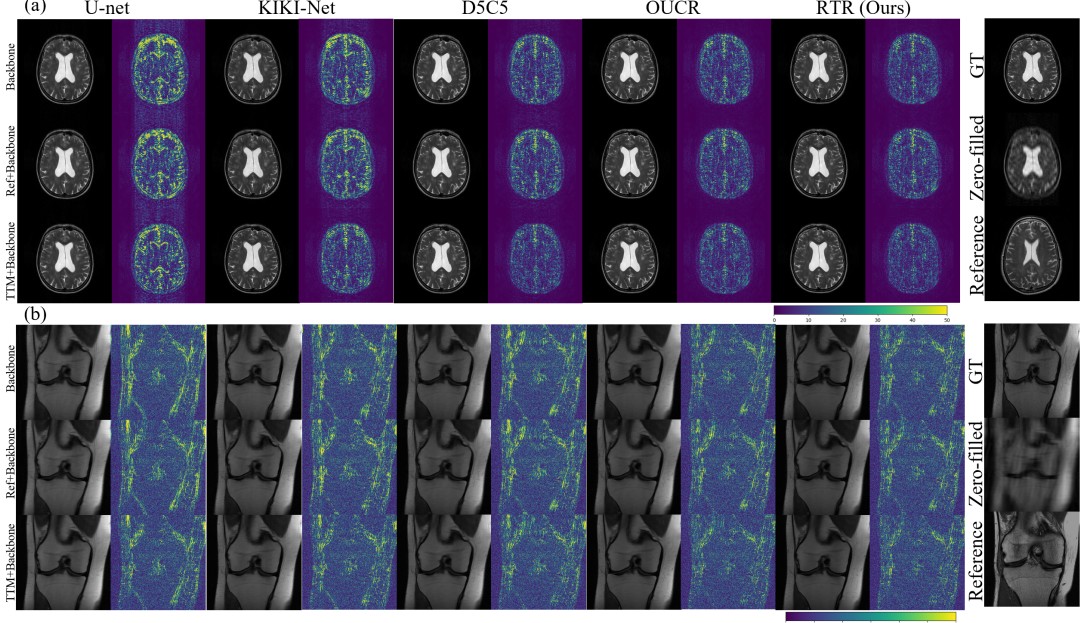

Figure 5: Visual comparison of different MRI reconstruction methods (AF=4) on (a) IXI and (b) fastMRI among 3 strategies. The second column of each sub-figure shows the absolute difference between the reconstructed image and the ground truth.

## Appendix E.  Additional Qualitative Comparison.

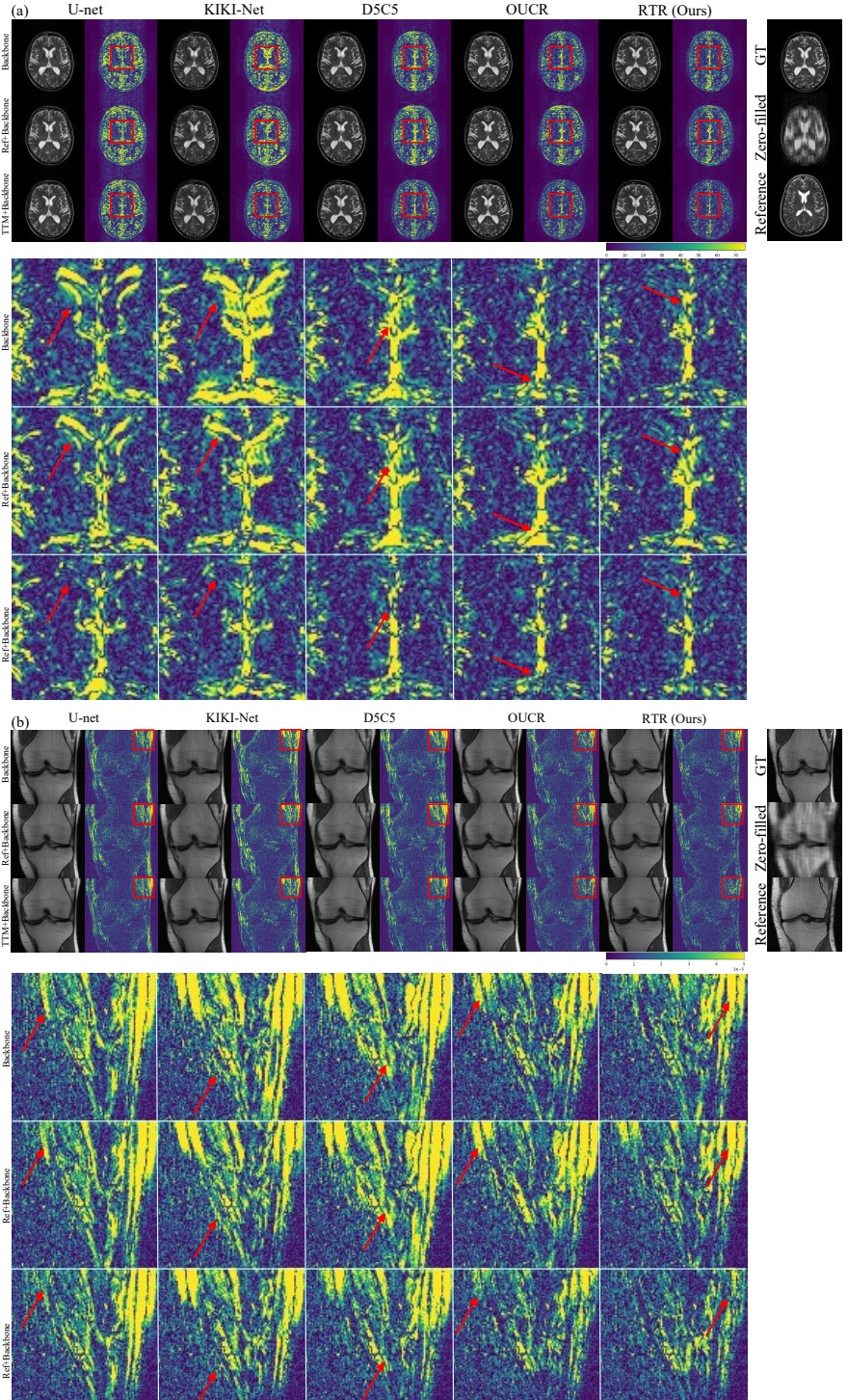

Figure 6: Visual comparison of different MRI reconstruction methods (AF=8) on (a) IXI and (b) fastMRI among 3 strategies. The red boxes of each subplot indicate where we zoom in on error maps. The red arrows point to where TTM obviously improves reconstruction quality.

## Appendix F. Discussion about Limitations.

While our proposed method yielded a competitive performance, there are potential areas for improvement. First, we are aware that our evaluation only relies on the image quality metrics (*i.e.*, SSIM and PSNR). PSNR provides a simple measure of the mean squared error between the original and reconstructed images and SSIM provides a more sophisticated measure of the structural similarity between the images that take into account human visual perception, but the proposed method can be potentially improved by incorporating more advanced image quality metrics, such as LPIPS (Zhang et al., 2018) and FID (Heusel et al., 2017) in training. Second, we use the reconstruction of CS MR data alone to demonstrate the performance of the proposed method. The attention mechanism of the proposed TTM can be generally applied to other imaging modalities, like under-sampled Ultrasound data and corrupted data from computer tomography, as mentioned by the reviewer. The goal of TTM is to discover and transfer correspondent features between the reference and input. We plan to include the application to other modalities in future work. Third, the performance of TTM can be compromised by the out-of-distribution input. While there is a slight performance drop compared to using Level 1 (L1) reference data as shown in Table 7, using "Abnormal" reference data can still improve reconstruction quality. In particular, OUCR and RTR are improved from 37.41 and 38.16 to 38.11 and 38.85, respectively. This is mainly because the proposed TTM is trained to discover and transfer correspondent features between the reference and under-sampled data. It is worth noting that we did not retrain TTM to adapt "Abnormal" reference data, the performance can be further improved by including "Abnormal" reference data during model training.

