# OpenReview forum: "Reference-based MRI Reconstruction Using Texture Transformer"
_MIDL.io/2023/Conference — MIDL 2023 Poster_

### Official Review · Reviewer_3LYK · 2023-01-30

**Confidence:** 5
**Preliminary Rating:** 2

**Summary:**

The paper proposes a Texture Transformer architecture for MRI reconstruction from undersampled acquisitions. The proposed method uses a reference MRI that can come from a different subject than the currently reconstructed image. The proposed method and the fact it uses a reference MRI from another subject are the main novelties of the paper.

The authors perform experiments on the IXI and the fastMRI datasets. They compare their methods against four other methods published in the literature. Finally, the authors present ablation studies to confirm that the proposed improvements are responsible for their improved MRI reconstruction results.

The paper's significance is investigating ways to improve MRI reconstruction, which would allow MRI scans to be faster.


**Strengths:**

The paper is mostly well-written, and the proposed method is relatively easy to follow. The authors intend to make the code and the pre-trained models publicly available. The method is compared with four other previously published works. The authors present many ablation studies that support the proposed improvements.

**Weaknesses:**

Literature review
The authors grouped the MRI reconstruction methods in two: single image reconstruction (SIR) and guidance-based image reconstruction (GIR). For GIR, the authors say that this type of method needs a fully-sampled auxiliary MR sequence to produce visually pleasing results. This is not always the case, for example, in multi-sequence MRI reconstruction [1,2]. There are also image translation methods [3].

Methods
It was unclear how the matched references are produced/chosen in the methods. This seems a crucial step in the methodology.  Is there an image registration step involved?

Experiments
- The IXI dataset does not have raw MRI data, only reconstructed data, which is sub-optimal for MRI reconstruction experiments. Was any action taken to make the IXI experiment more realistic?
- The fastMRI dataset serves as a benchmark: https://fastmri.org/ Why not use the same data split, so the experiments of the authors be directly comparable with the methods submitted to the benchmark?
- The methods compared are relatively old or are not state-of-the-art (cf. [4,5]). Why not compare against state-of-the-art, such as https://github.com/NKI-AI/direct?

Results
- I missed discussions about the quality of the results, other than the metrics of the proposed method are better. For example, in Figure 3, where should the reader pay attention to?

- For the ref + backbone results, the authors claim that there was almost no improvement in the results. There is literature pointing otherwise [6,7], but registering the images is critical, and it was unclear to me whether this step was taken.

[1] Peng, C., et al. “Towards multi-sequence MR image recovery from undersampled k-space data.” MIDL 2020.
[2] Xiang, L., et al. “Deep-Learning-Based Multi-Modal Fusion for Fast MR Reconstruction.” IEEE Trans. Biomed. Eng. 66.7 (2018): 2105-2114.
[3] Welander, P., Karlsson, S. and Eklund, A., 2018. Generative adversarial networks for image-to-image translation on multi-contrast mr images-a comparison of cyclegan and unit. arXiv preprint arXiv:1806.07777.
[4] Muckley, M.J., Riemenschneider, B., Radmanesh, A., Kim, S., Jeong, G., Ko, J., Jun, Y., Shin, H., Hwang, D., Mostapha, M. and Arberet, S., 2021. Results of the 2020 fastMRI challenge for machine learning MR image reconstruction. IEEE transactions on medical imaging, 40(9), pp.2306-2317.
[5] Beauferris, Y., Teuwen, J., Karkalousos, D., Moriakov, N., Caan, M., Rodrigues, L., Lopes, A., Pedrini, H., Rittner, L., Dannecker, M. and Studenyak, V., 2020. Multi-channel MR Reconstruction (MC-MRRec) Challenge--Comparing Accelerated MR Reconstruction Models and Assessing Their Genereralizability to Datasets Collected with Different Coils. arXiv preprint arXiv:2011.07952.
[6] Souza, R., Beauferris, Y., Loos, W., Lebel, R.M. and Frayne, R., 2020. Enhanced deep-learning-based magnetic resonance image reconstruction by leveraging prior subject-specific brain imaging: Proof-of-concept using a cohort of presumed normal subjects. IEEE Journal of Selected Topics in Signal Processing, 14(6), pp.1126-1136.
[7] Beauferris, Y., Lasby, M. and Souza, R., 2022, March. Leveraging Multi-Visit Information for Magnetic Resonance Image Reconstruction: Pilot Study on a Cohort of Glioblastoma Subjects. In 2022 IEEE 19th International Symposium on Biomedical Imaging (ISBI) (pp. 1-5). IEEE.

**Deanonymize Review:**

no

**Detailed Comments:**

Overall, I think the idea is very promising, but the experiments and the methods used in the comparison are limited. My main comments are in the weaknesses section of my report.

.



**Paper Type:**

methodological development

**Questions To Address In The Rebuttal:**

I would like the reviewers to address all the points I listed in the weaknesses section. Many of them are easy, such as clarifying methodological details, improving the literature review, and improving the discussion of the results.

My major concerns are the experiments, which in my opinion, could have been better designed:
- The IXI dataset is not an appropriate dataset for MRI reconstruction experiments
- The four methods used in the comparison do not represent the current state-of-the-art for MRI reconstruction
- As I mentioned, I have seen other works in the literature [6,7] that are very similar to the ref + backbone model presented in the paper, I am quite surprised that the results comparing backbone versus ref + backbone are so similar. Is the reference registered to the zero-filled reconstruction in the ref + backbone experiment?

---

### Official Review · Reviewer_k2af · 2023-02-03

**Confidence:** 4
**Preliminary Rating:** 4
**Recommendation:** Poster

**Summary:**

The manuscript "Reference-based MRI Reconstruction Using Texture Transformer" presents a extended and (partly) novel approach to reconstructed magnetic resonance imaging (MRI) data that were undersampled during compressed-sensing (CS) acquisition. The presented approach is two-fold. On one hand a texture transformer module (TTM) is presented that can be attached before existing reconstruction solutions. On the other hand, a recurrent transformer reconstruction backbone (RTR) solution is presented which unifies the processing of the TTM and an image reconstruction algorithm.
Overall, the work sounds solid while there might be some limitations of the authors in the field of MRI, their knowledge in deep learning sounds very profound. The modular character of the presented work is very interesting. The introduction motivates the work and gives context. The Methodology section goes into details of the neural networks. The section 3 Experiments and Results summarizes information on the datasets and reference methods as well as the results. A rather short conclusion statement ends the manuscript.

In my opinion, the details in the context of DL are very profound and have value for the scientific community. The results have not fully convinced me, the predicted images do - from my perspective - not clearly outperform existing and partly easier methods, eventough the reported quantification results underline the value of a TTM and also show that the RTR-reconstructions are best performing. I do like the idea of TTM which might eliminate or at least reduce the need of pairly auxiliary sequences as reference for reconstruction. To the best of my knowledge this (use of paired auxiliary data) is however not something that has gained wide attention in the (MRI) community and if so, matched reference data (e.g. a T1) are acquired in a full clinical protocol anyways.

**Strengths:**

- The presented work convinces me with respect to the presented developments and details in neural network architecture and data management strategy. In addition, the basic idea sounds appealing. The comparison with other methods is done very fair and solid.
- The methodological aspects of this work are of potential interest of the MIDL community and show timely approaches for image processing.
- It was pointed out that finding a matched reference is difficult, but (weakness) in particular in extreme cases (very different anatomy) this might be a even larger or absolute problem.
- High level of novelty (modular expansion/improvement of existing recon algorithms with TTM)

**Weaknesses:**

- 3. Experiments and Results, Reconstructions Results: the statements of outperforming are to strong in my opinion. The overall benefit / gain in reconstruction performance is not overwhelming.
- 3. Experiments and Results, Reconstructions Results: "... all models that make use of TTM exhibit better ability of suppressing overall errors" This seems to be supported by the numbers (PSNR/SSIM), but I am not totally convienced by my visual impression of the images in Fig. 3. Please highlight (e.g. arrows) reconstruction errors in the other models. The difference maps appear also fairly structured such that it may reflect mainly slight variations in absolute image grayvalues rather then structure or contrast.
- 4. Conclusion: No comment on potential performance on real-world data is given. Simulated CS-data are not identical to retrospectively simulated ones. It is obvious that availablity of real data is limited and might be difficult. But addressing this or ideally including an example of the performance on real data form another source would be very interesting and strengthen the work.
- 4. Conclusion: There is no discussion and in particular no limitations are named.


**Deanonymize Review:**

yes

**Detailed Comments:**

- 1. Introduction: Compresses Sensing first publication was (to the best of my knowledge): Lustig, M., Donoho, D. L., Santos, J. M., & Pauly, J. M. (2008). Compressed Sensing MRI. IEEE Signal Processing Magazine, 25(March 2008), 72–82. https://doi.org/10.1109/MSP.2007.914728
- 2. Methodology, Eq. 1: I would state that Eq. 1 rather descibes a Fourier + Inv. Fourier Transform incl. a sampling pattern and added noise to a ground truth. Are x and y images? Typically the data acqusition is formulated with the so-called forward model, e.g. y = Ax +epsylon
x = image/object
y = measured data, aka MR signal
A = UFC Y -> operators such as C = receiver coil sensitivity, F = Fourier Transform, U = (under-) sampling matrix.
epsylon = noise. If you refer with Eq 1 to Eq1 in Schlemper 2017 "A Deep Cascade of Convolutional Neural Networks for MR Image Reconstruction" then x and y are in different domains. Of course, rewriting this is absolutly fine, but in particular undersampling is normally considered in k-space domain as this represents the data acuqisition domain.
- 3. Experiments and Results, Reconstructions Results: Ordering of figures. Fig. 4 mentioned before Fig. 3.


**Paper Type:**

methodological development

**Questions To Address In The Rebuttal:**

Our work sounds overall solid, but the interpretation of the results are not well balanced. The reported "outperformance" seems not so clear to me. Could you either add results that show the clear benefit of your work, e.g. more clear increase in SSIM and/or images where you could point the reader to reconstruction errors that are removed by your proposed method (either by adding TTM or performing the reconstruction wtih RTR)?
Overall, to the best of my knowledge, your work is interesting and relevant from a methodological / technological point of view. But the problem you address (reconstruction of CS MR-data) seems not to perfectly fit this. E.g. if you could show that your approach is also beneficial for undersampled Ultrasoud data or corrupted data from computer tomography, you could strengthen your method by detaching ot from CS-MRI alone.

---

### Official Review · Reviewer_KG6M · 2023-02-05

**Confidence:** 4
**Preliminary Rating:** 4
**Recommendation:** Poster

**Summary:**

This paper introduces a new MRI reconstruction approach for under-sampled MRI that takes advantage of an unpaired fully-sampled reference image of the same modality to improve the texture quality. Alongside this development, a recurrent transformer for MRI reconstruction is introduced. Both of these advances are evaluated on IXI (brain) and fastMRI (knee) and show an improvement in performance in terms of PSNR and SSIM.

**Strengths:**

This paper introduces a new model and approach for utilising unpaired fully-sampled reference images for improving MRI reconstruction. The results show a promising improvement in PSNR and SSIM. The boxplots provide a useful visualisation of the variability across images, although they are a little small. The description of the model is generally pretty clear, although a little dense at points. The evaluations appear to be fair comparisons, with several test conducted, and a detailed ablation study is performed to show the effect of each change.

**Weaknesses:**

The forward model in equation 1 implies that the data representation for the observations is in image space rather than k-space. Such a formulation seems problematic for considering under-sampled k-space data, as unobserved frequencies manifest as blurry image reconstructions rather than missing data. It's not clear if the model takes complex data as input (magnitude + phase? or i and j) and I'm uncertain if the IXI dataset provides a reasonable test case for MRI reconstructions, as I don't believe the phase information is available? The loss function is not clearly defined, it's just stated as L1 - is that on the complex signal at each pixel?

In terms of the validation, the number of experiments is commendable but it would be good to have some discussions of the employed metrics. there is no discussion of the difference in meaning of PSNR and SSIM. The paper is also about texture, and PSNR is invariant to texture as it's a purely voxel/pixelwise measure so it doesn't backup the texture aspect. SSIM provides some description of texture, but it's poorly correlated with human notions of texture (see LPIPS for example). It would be helpful to highlight some area of interest in Fig 3.  as it stands it's hard to know what to focus on.

One thing that is not explored or discussed is the influence of the reference image in situations where either the under-sampled input, or the reference image, is drawn from an abnormal distribution (e.g. a disease population). What is the expected behaviour?

Std-devs should be included for numbers in tables.

**Deanonymize Review:**

no

**Paper Type:**

methodological development

**Questions To Address In The Rebuttal:**

A discussion/explanation of the forward model is required as well as some discussion about evaluation metrics. At least a discussion, but ideally a provisional result on a toy example regarding abnormal data would be beneficial. Perhaps some of the less important architecture details can be relegated to the appendix to make space.

---

### Meta-Review · Area_Chair_zFLg · 2023-02-23

**Recommendation:** Accept (Poster)
**Confidence:** 5

**Metareview:**

All reviewers recommended acceptance of this manuscript after authors' rebuttal. They agreed that the proposed method is new and will be of a good interest to MIDL community. The authors shall carefully address all questions and concerns (as discussed in the rebuttal) in a revised version.